# Comparative Transcriptomic and Metabolomic Analyses Reveal the Regulatory Effect and Mechanism of Tea Extracts on the Biosynthesis of *Monascus* Pigments

**DOI:** 10.3390/foods11203159

**Published:** 2022-10-11

**Authors:** Wen-Long Li, Jia-Li Hong, Jin-Qiang Lu, Shan-Gong Tong, Li Ni, Bin Liu, Xu-Cong Lv

**Affiliations:** 1Food Nutrition and Health Research Center, School of Advanced Manufacturing, Fuzhou University, Jinjiang 362200, China; 2College of Food Science, Fujian Agriculture and Forestry University, Fuzhou 350002, China; 3Institute of Food Science and Technology, College of Biological Science and Technology, Fuzhou University, Fuzhou 350108, China

**Keywords:** *Monascus* pigments, tea extracts, transcriptomics, metabolomics, RT-qPCR

## Abstract

*Monascus* pigments (MPs) are natural edible pigments with high safety and strong function, which have been widely used in food and health products. In this study, different types of tea extracts (rich in polyphenols) were used to regulate the biosynthesis of MPs. The results showed that 15% ethanol extract of pu-erh tea (T11) could significantly increase MPs production in liquid fermentation of *Monaco’s purpureus* M3. Comparative transcriptomic and metabolomic analyses combined with reverse transcription-quantitative polymerase chain reaction (RT-qPCR) were used to further explore the regulatory mechanism of T11 on the biosynthesis of MPs. Comparative transcriptomic analysis showed that there were 1503 differentially expressed genes (DEGs) between the Con group and the T11 group, which were mainly distributed in carbohydrate metabolism, amino acid metabolism, energy metabolism, lipid metabolism, metabolism of terpenoids and polyketides, etc. A total of 115 differential metabolites (DMs) identified by metabolomics between the Con and T11 groups were mainly enriched in glutathione metabolism, starch and sucrose metabolism, alanine, aspartic acid and glutamate metabolism and glycine, serine and threonine metabolism, etc. The results of metabolomics were basically consistent with those of gene transcriptomics, indicating that the regulatory effect of T11 on the biosynthesis of MPs is mainly achieved through affecting the primary metabolic pathway, providing sufficient energy and more biosynthetic precursors for secondary metabolism. In this study, tea extracts with low economic value and easy access were used as promoters of MPs biosynthesis, which may be conducive to the application of MPs in large-scale industrial production. At the same time, a more systematic understanding of the molecular regulatory mechanism of *Monascus* metabolism was obtained through multi-omics analysis.

## 1. Introduction

*Monascus* pigments (MPs) are natural edible pigments which have been widely utilized in the processing of food and health products in Southeast Asia. Due to MPs’ superior safety performance, several animal experiments have demonstrated that there are no acute or chronic poisoning and no mutagenic effects after consuming foods containing MPs over an extended period of time [1,2]. In addition, more and more studies have shown that MPs have a variety of biological activities, such as anti-inflammatory [1], anti-cancer [3] and anti-hyperlipidemia [4]. It is generally believed that the biosynthesis of MPs is mainly realized through the fatty acid synthase (FAS) pathway and polyketide synthase (PKS) pathway [5,6]. The market demand of MPs is increasing year by year with an annual growth rate of 5–10% as people gradually realize the excellent food processing uses and bioactive functions of MPs [7]. However, there is no good chemical method to produce MPs, which are mainly produced by biological fermentation. Therefore, improving the production of MPs in liquid state fermentation is an important topic of *Monascus* research.

Both strain mutation and gene modification can significantly increase the production of MPs by *Monascus* spp. [8,9]. However, due to the food safety of genetically engineered products, the social acceptance of genome editing technology is still at a low level, which hinders the promotion and application of genetically engineered *Monascus* strains. Adjusting fermentation parameters is one of the most important ways to improve the quality and production of MPs. Optimization of culture conditions is still a safe, economical and effective method, which can be divided into physical and chemical regulation methods. In recent years, many studies have been conducted to improve the production of MPs by regulating temperature [10], light [11], magnetic field [12] and surfactant extraction [13] and other physical methods. The chemical regulation methods mainly include adjusting the nutrient composition of the culture medium and adding exogenous compounds. For example, the addition of exogenous acetate species could transfigure the morphology and biomass of *Monascus ruber* M7 and enhance its MPs production [14]. Zhen et al. [15] found that adding 0.02 M NaCl to the medium could significantly increase the production of MPs and monacolin K by *Monascus purpureus* SKY219. More and more studies have shown that tea extract is rich in polyphenols, which have various biological activities, such as antioxidant [16], anti-cancer [16], anti-inflammatory [17] and anti-hyperlipidemia activities [18]. Previous studies have shown that the addition of some polyphenols and tea extracts to the fermentation medium could significantly reduce the production of mycotoxins [19,20,21]. However, few studies focus on the regulatory effect and mechanism of tea extracts on the biosynthesis of MPs.

Omics technology can reveal the metabolic regulation mechanism from multiple perspectives, and has been widely used in the metabolic research of *Monascus* spp. In a previous study, comparative transcriptomic analysis was applied to analyze the effects of ATP-citrate lyase over-expression on pigment production in *M. ruber* [22]. The results showed that over-expression of ATP-citrate lyase regulated the expression of genes related to glycolysis, amino acid metabolism and pigment synthesis, thus promoting MPs production. In another case, differentially expressed genes (DEGs) derived from comparative transcriptomic showed that NH_4_Cl and NH_4_NO_3_ could up-regulate the expression of most genes related to carbon and nitrogen metabolism and amino acid biosynthesis in liquid fermentation of *M. purpureus*, and generate more precursors for MPs biosynthesis to increase MPs production [23]. In addition to comparative transcriptomic analysis, metabolomic analysis is also frequently used to study the metabolic regulation mechanism of *Monascus* spp. For example, disruption of the *pksCT* gene in *M. aurantiacus* resulted in obvious changes in the production of pigment and citrinin. Meanwhile, 18 differential metabolites (DMs) involved in acetyl-CoA, amino acid metabolism and citric acid metabolism were identified by comparative metabolomic analysis, which can provide important insights into the metabolic pathways affected by the *pksCT* gene in *M. aurantiacus* [24]. However, the biomolecular changes revealed by single omics are difficult to meet the needs of analyzing the metabolic mechanism. Therefore, in this study, comparative transcriptomic and metabolomic analyses were used to help us more systematically analyze the molecular mechanism of MPs anabolism induced by tea extracts from both the genes and metabolites aspects.

In this study, 12 types of easily available and economical tea extracts (rich in polyphenols) were used as a regulatory factor to investigate the effect on the biosynthesis of MPs in this study. Then, after the optimal type of tea extract was selected, the regulatory mechanism of *M. purpureus* M3 liquid fermentation was explored through comparative transcriptomic and metabolomic analyses combined with RT-qPCR. This study will help us to understand the molecular regulation mechanism of MPs biosynthesis more systematically and comprehensively, and lay a potential application foundation for the industrial-scale production of MPs.

## 2. Materials and Methods

### 2.1. Strain and Materials

*Monascus purpureus* M3, obtained from red mold rice of Fujian Province in China, and stored in Food Biotechnology Laboratory, Fujian Agriculture and Forestry University (Fuzhou, China). Strain activation was carried out on potato dextrose agar (PDA) (Guangdong Huankai Microbial Sci. & Tech. Co., Ltd., Guangzhou, China), incubated at 28 °C for 7 days, and then stored at 4 °C. Green tea, black tea, white tea and pu-erh tea were purchased from the local Yonghui Supermarket (Fuzhou, China).

### 2.2. Preparation and Component Identification of Tea Extracts

Four different kinds of tea were extracted with water, 15% ethanol and 75% ethanol at room temperature, respectively (Table 1). The extracts were centrifuged at 3500× *g* for 10 min at room temperature. Then, the supernatant was concentrated and then freeze-dried. All tea extract samples were stored at −20 °C for subsequent use. Total polyphenol content was measured by Folin–Ciocalteau reagent. The standard curves were used to convert the average absorbance of each sample into milligrams per gram gallic acid equivalent. The test methods of tea extracts are detailed in (Appendix A).

### 2.3. Measurement of MPs Production

The fermentation broth and mycelium in single well of the 24 micro-well plate were transferred to 1.5 mL centrifuge tube and dried at 60 °C. Then, extracted with 1 mL 70% ethanol in 200 W ultrasonic bath for 30 min at 55 °C. The extract was centrifuged at 8000 rpm and 25 °C for 5 min. The absorbance values (OD) of the supernatant (diluted in 70% ethanol) were detected at 390, 465 and 510 nm by SpectraMaxi3x microplate reader (Molecular Devices, San Francisco, CA, USA) to measure the production of yellow, orange and red pigments, respectively. Single color value (OD) = (OD390 nm, OD465 nm or OD510 nm) × dilution times. Total color value (OD) = (OD390 nm + OD465 nm + OD510 nm) × dilution times.

### 2.4. Preparation of M. purpureus M3 Spore Suspension

*M. purpureus* M3 activation was carried out on sterile PDA slants and incubated in a constant temperature incubator at 30 °C for 7 days. After a large number of spores were formed on the medium, aseptic spore washing solution was added to the Petri dishes, and the mycelium was filtered with sterile gauze to remove the culture medium and other impurities. Then, the spore concentration was adjusted to 10^6^ spore/mL with aseptic water for subsequent 24 micro-well plate fermentation.

### 2.5. Effect of Tea Extract Addition on MPs Production

*M. purpureus* M3 was fermented on a 24 micro-well plate, and each well contained 1 mL of sterile liquid medium or tea extract medium. Then, 10% (*v/v*) spore suspension was inoculated into each well and cultured in an incubator at 30 °C for 10 days. The composition of the liquid fermentation medium was as follows: 5% glucose, 1% peptone, 0.25% KH_2_PO_4_, 0.01% MgSO_4_·7H_2_O and 0.02% chloramphenicol dissolved in deionized water. Meanwhile, 12 tea extracts (T1-T12) with 3 different total polyphenol concentrations (H: 250 μg/mL, M: 125 μg/mL and L: 62.5 μg/mL) were added into above medium to prepare tea extract medium, respectively. The culture medium of fermentation endpoint was collected and used for MPs determination to determine the effect of different tea extracts on MPs production.

### 2.6. Effects of the Concentration of Tea Extracts and the Fermentation Time on MPs Production

The optimal tea extract from T1–T12 was selected as the supplement of *M. purpureus* M3 fermentation to determine the optimal concentration of tea extract to promote MPs production. On the basis of the above-mentioned liquid fermentation medium, the optimal tea extracts of 10 gradient total polyphenol concentrations (25 μg/mL, 50 μg/mL, 75 μg/mL, 100 μg/mL, 125 μg/mL, 150 μg/mL, 175 μg/mL, 200 μg/mL, 225 μg/mL and 250 μg/mL) were added, respectively, for *M. purpureus* M3 microplate fermentation. The spore suspension (10%, *v/v*) was inoculated into 24 micro-well plate with 1 mL of the aseptic fermentation media per well and cultured in an incubator at 30 °C for 10 days. The culture medium of fermentation endpoint was collected and used for MPs determination to determine the effect of different concentration of tea extracts on MPs production.

After the optimal supplemental concentration of tea extract was selected, the optimal concentration of tea extract was used as the supplement of culture medium to study the effect of fermentation time on MPs production. The spore suspension (10%, *v/v*) was inoculated into 24 micro-well plate with 1 mL of the aseptic fermentation media per well and cultured in an incubator at 30 °C for 20 days. Fermentation broth and mycelium were collected every two days to explore the effect of tea extracts on the MPs fermentation process.

### 2.7. RNA Extraction Library Construction and Sequencing

The harvested *Monascus* mycelia were immediately frozen in liquid nitrogen after each sampling and stored at −80 °C until the RNA extraction. Total RNA of *Monascus* mycelium in fermentation medium was isolated and extracted using a Spin Column Fungal Total RNA Purification Kit (Sangon Biotech, Shanghai, China), and then used for cDNA library construction and high-throughput sequencing at Majorbio Technology Corporation (Shanghai, China) on the Illumina HiSeq™ 4000 sequencing platform (Illumina™, San Diego, CA, USA).

### 2.8. Transcriptome Assembly, Mapping

The clean reads were obtained by removing reads containing adapter, reads containing ploy-N and low-quality reads from raw data. The clean reads were mapped to the whole genome of *M. purpureus* BL3 we have sequenced as the reference genome by using HISAT2 (daehwankimlab.github.io/hisat2/; accessed on 12 March 2020). At the same time, the Q20, Q30 and GC content of the sequencing results were calculated to measure the quality of the sequencing results. In addition, the fragments per kilobase of transcript per million fragments mapped (FPKM) or the transcripts per kilobase of per million mapped reads (TPM) was used to measure the level of gene expression [25]. Genes with expression levels (log_2_^(TPM+1)^) greater than 1 were used to draw Venn charts and bar charts.

### 2.9. Gene Differential Expression Analysis and Functional Annotation

Differential gene expression analysis between the two groups was performed by DESeq R software package (ver. 1.10.1, Simon Anders, Heidelberg, Germany). DESeq provided a model based on the negative binomial distribution to determine differential expression in digital gene expression data. As a result of the Benjamini–Hochberg approach for controlling the false discovery rate, the genes with adjusted *p*-value < 0.01 detected by DESeq were considered as differentially expressed genes (DEGs). The GOG (https://www.ncbi.nlm.nih.gov/COG/; accessed on 16 March 2020) and KEGG (http://www.genome.jp/kegg/; accessed on 16 March 2020) databases were used for functional annotation and enrichment analysis of DEGs. Among them, KEGG is a biological systems database, which can be used to infer the system behavior of organisms by linking transcriptomes to the environment and analyzing metabolic pathways [26]. An analysis of KEGG pathways enriched for differential expression genes was conducted using KOBAS software [27].

### 2.10. RT-qPCR Analysis of MPs-Biosynthesis-Related Genes

The total RNA was extracted by Spin Column Fungal Total RNA Purification Kit, and then the cDNA was obtained by reverse transcription performed in a metal bath with PrimeScript™ RT Reagent Kit with gDNA Eraser Kit (Takara, Dalian, China). We stored the cDNA in a refrigerator at −80 °C. Finally, the cDNA was treated with SYBR Premix Ex Taq™ II (TLI RNaseH plus) kit (Takara, Dalian, China). The cDNA was detected by real-time fluorescent quantitative PCR in ABI 7300 instrument (Applied Biosystems, Waltham, MA, USA). The relative expression of MPs synthesis gene was analyzed. The preparation of amplification system and operation process refer to the manual of the kit. The primers used are shown in (Appendix A) [28,29]. The relative levels of target mRNAs were determined using the 2^−∆∆Ct^ method and were normalized to the β-actin mRNA signals in each sample.

### 2.11. Metabolomic Analysis of Fermentative Mycelium

Six mycelium samples were taken from the Con group and T11 group, respectively. A combination of 30 ± 1 mg of each sample, 450 μL of extraction solution (methanol: chloroform = 3:1) and 10 μL L-2-chlorophenylalanine were mixed by vortex for 30 s. Then, we added ceramic beads and treated at 45 Hz for 4 min by grinder (JXFSTPRP-24, Jinxin, Shanghai, China). Ultrasonic for 5 min (ice water bath) and centrifuged the sample at 12,000 rpm, 4 °C for 15 min. Transferred supernatant into 1.5 mL centrifuge tube, 20 μL of each sample were mixed to obtain quality control (QC) sample. The extract was dried in a vacuum concentrator and 40% methoxyamine salt reagent (methoxyamine hydrochloride, soluble in pyridine 20 mg/mL) was added to the dried metabolite. Next, 60 μL BSTFA (containing 1% TMCS, *v/v)* was added to each sample, and the mixture was incubated at 70 °C for 1.5 h. After cooling to room temperature, 5 μL FAMEs (dissolved in chloroform) was added to the mixed sample.

Gas chromatography-time-of-flight mass spectrometry (GC-TOF-MS) analysis was carried out by Agilent 7890B gas chromatography system (Agilent, Santa Clara, CA, USA) coupled with a PEGASUS HT mass spectrometer (LECO, St. Joseph, MI, USA). Each sample (1 μL) was separated by DB-5MS capillary column (30 m × 250 μm × 0.25 μm, Agilent, Santa Clara, CA, USA) coated with 5% diphenyl cross-linked with 95% dimethylpolysiloxane (Agilent, Santa Clara, CA, USA). The specific analysis conditions of GC-TOF-MS mainly referred to Huang et al. [24], with some modifications as follows: helium (purity of 99.9999%) gas flow rates of septum purge and through the column: 3 and 1 mL/min, respectively. Initial temperature of 50 °C for 1 min, increased to 310 °C at a rate of 10 °C/min. The temperatures of forward inlet, ion source and transfer line were 280 °C, 280 °C and 250 °C, respectively. Ions were generated by 70 eV electron ionization (EI). After a solvent delay of 6.03 min, the EI spectra were acquired in the range of m/z 50–500 at a rate of 12.5 spectra/s. Each sample was analyzed with three independent repeats.

### 2.12. Statistical Analysis

All results were performed in triplicate and presented as the means ± standard deviation (SD). Statistical differences between multiple samples were determined by one-way ANOVA and *t*-test in GraphPad Prism software (ver. 7.0, GraphPad Inc., San Diego, CA, USA).

## 3. Results

### 3.1. Effects of Different Kinds of Tea Extracts on MPs Production

In this study, 12 kinds of tea extracts were added to the fermentation medium of *Monascus* to explore their effects on the production of MPs. After 10 days of fermentation, there were obvious differences in MPs production among the different experimental groups (Figure 1). When T2, T3, T5, T6, T9 and T12 were added to the fermentation medium, the biosynthesis of MPs was inhibited to some extent, no matter how much they were added (*p* < 0.05). Interestingly, T10 and T11 could promote the production of MPs at high, medium and low concentrations (*p* < 0.01), and the regulation effect of high concentration of T11 was better. Compared with the control group, the production of yellow, orange and red pigments in the T11-H group increased by 174.85%, 235.02% and 172.14%, respectively, which were higher than other experimental groups, indicating that T11 could be used as a fermentation supplement to promote MPs biosynthesis during *M.*
*purpureus* M3 fermentation.

### 3.2. Composition Analysis of Different Tea Extracts

In order to explore the potential reasons for the different regulation effects of tea extracts on MPs synthesis, LC-MS was used to analyze the compound composition of different tea extracts. A total of 37 polyphenols were identified in 12 tea extracts, including epicatechin (EC), catechin, (-)-epigallocatechin (EGC), catechin gallate, epigallocatechin gallate (EGCG), quercetin, rutin, etc. (Figure 2A). Tea extracts (T2, T5, T8) treated with low-concentration ethanol (15%) showed higher levels of catechin, rhoifolin and reynoutrin, etc., than other treatment groups. In the green tea extracts (T1, T2, T3), several primary catechins, such as EG, EGCG and EGC, are obviously higher than other types of tea extracts. Meanwhile, principal component analysis (PCA) and hierarchical cluster analysis (HCA) were also carried out to analyze the similarities and differences of different tea extracts. The PCA score plot showed that the projection points of tea extracts (T10, T11, T12) derived from pu-erh tea are relatively close (Figure 2A), where they were mainly characterized by mauritianin, ellagic acid, catechin 7-*O*-apiofuranoside and naringenin, which indicates the similarity of their components (Figure 2B,C). The projection points of tea extracts treated with 75% alcohol are also close to each other, which indicated that different treatments would affect the composition of substances (Figure 2B,C). Correspondingly, the result of the HCA visually revealed that the compounds profiles of 12 tea extracts could be classified into four major clusters, namely cluster I (T4), cluster II (T1, T2, T5, T7, T8), cluster III (T10, T11, T12) and cluster IV (T3, T6, T9), indicating that the composition of tea extracts might show obvious differences due to different sources and treatments (Appendix A). Compared with other tea extracts with poor or negative effects on MPs production, the main substance characteristics of T11 are high concentrations of ellagic acid, taxifolin and chlorogenic acid, which may be potential factors for T11 to promote MPs production (Figure 2A). Of course, another possible explanation is that T11 contains fewer substances that inhibit MPs production, whereas these substances may be more abundant in other tea extracts.

### 3.3. Effects of Tea Extracts Concentration and Fermentation Time on MPs Production

The effect of various concentrations of T11 on MPs production is shown in Figure 3. Compared with the Control group without treatment, low concentrations of T11 (25 μg/mL, 50 μg/mL) have no significant enhancement effect on MPs production. When the supplemental concentration of T11 was higher than 75 μg/mL, the MPs production increased significantly. Notably, the most significant increase in MPs production was observed at a T11 concentration of 175 μg/mL. However, with the increase in T11 concentration from 175 μg/mL to 250 μg/mL, the promotion rate of MPs production decreased. Therefore, 175 μg/mL was used as the optimal supplemental concentration of tea extract T11 to further investigate the effect of fermentation time on MPs production.

In order to investigate the effect of fermentation time on the production of MPs at the optimal supplemented concentration of T11, we carried out fermentation for 20 days. During this period, fermentation broth and mycelium were collected every 2 days for the determination of MPs production (Figure 4). The production of MPs in both the Con group and T11 group was at a low level in the first 8 days without significant difference. After the 8th day, MPs production increased significantly, and there was a significant difference between the experimental group and the control group. The MPs production of the T11 group was significantly higher than that in the Con group, and the growth rate production reached the maximum on the 14th day. After the 18th day of fermentation, the production of MPs gradually tended to be stable in the T11 group, indicating that the production capacity of MPs gradually weakened, and the fermentation basically ended. The increase rate of pigment production in the Con group reached the maximum on the 16th day, and then the MPs production gradually stabilized. In summary, T11 can be a supplement for *M. purpureus* M3 liquid fermentation to promote MPs production. The growth rate of MPs reached the peak on the 14th day of fermentation. After 18 days, MPs production tended to be stable, and the total color value of the T11 group was 151.77 ± 5.30 U/mL, which was approximately four times that of the Con group. Considering that the fastest growth rate of MPs production was on the 14th day of fermentation, indicating that the metabolic activity of *M. purpureus* M3 was vigorous, mycelia were used for transcriptome sequencing to further introduce the regulatory mechanism of T11 on MPs synthesis.

### 3.4. High-Throughput Sequencing and De Novo Assembly

High-throughput transcription sequencing was helpful to better explore the regulatory effects of T11 on the biosynthetic pathways of MPs. The GC% of the sequencing data from six samples ranged from 52.27% to 53.56%, and the Q20 and Q30 was not less than 97.32% and 92.48%, respectively (Appendix A), which indicates that the data are qualified. According to the comparison results, there was no contamination in the experiment, since the comparison rate of reads and reference genomes of six samples was higher than 97%. The percentage of clean reads number with multiple alignment positions on the reference sequence was not more than 4%, which indicated that the sequencing data were sufficiently accurate and high quality to support further analysis [30]. Pearson’s correlation coefficient was used as the indicator of biological correlation. The results show that there was a strong correlation between the two groups of samples (*r* > 0.9), indicating that the two groups of samples have good repeatability (Appendix A). These results showed that selecting the genome of *M. purpureus* BL3 as the reference genome was appropriate.

### 3.5. Function Analysis and Annotation of DEGs

The transcriptome sequencing results were compared to analyze the regulatory effect of T11 on MPs biosynthesis. There were 7332 genes shared between the two groups (Figure 5A). DEGs were found between the two sample groups, and the differences and statistical significance of gene expression levels were shown by volcano plots (Figure 5B). Compared with the control group, the up-regulated and down-regulated DEGs of the T11 group were 513 and 990, respectively. Hierarchical cluster analysis of the screened DEGs was also performed in Figure 5C.

All genes obtained by transcriptome assembly were compared with six databases (NR, Swiss prot, Pfam, EggNOG, GO and KEGG) to discern the functional information of genes and make statistics on the annotation of each database (Appendix A). To further interpret the results of RNA-Seq data, the COG database was used to classify DEGs. The DEGs between Con and T11 were divided into 21 functional categories in the COG database (Figure 6), mainly including functional description (77), intracellular trafficking, secretion and vesicle transport (65), carbohydrate transport and metabolism (59), lipid transport and metabolism (48), amino acid transport and metabolism (47) and energy production and conversion (38), etc. According to the annotation results from the NR database (Appendix A), the genes related to MPs biosynthesis (such as “PKS1”, “fatty acid synthase alpha subunit” and “fatty acid synthase beta subunit”) were all up-regulated in the T11 group, indicating that T11 may increase MPs production by promoting the expression of genes involved in the MPs biosynthesis pathway.

The KEGG database was used to further analyze the metabolic pathway involved by DEGs. The DEGs annotated by KEGG are shown in Figure 7A, which were mainly distributed in metabolism including carbohydrate metabolism (74), amino acid metabolism (73), energy metabolism (34), lipid metabolism (31), metabolism of other amino acid (22), metabolism of cofactor and vitamins (17) and metabolism of terpenoids and polyketides (11) (Figure 7A). Specific to metabolic pathway, the significant difference of the top 20 metabolic pathways is shown by bubble diagram, including glycolysis/gluconeogenesis (20), tryptophan metabolism (16), pyruvate metabolism (14), arginine and proline metabolism (14), valine, leucine and isoleucine degradation (13), tyrosine metabolism (13), cysteine and methionine metabolism (13), phenylalanine metabolism (12), glyoxylate and dicarboxylate metabolism (12) and fatty acid degradation (11), etc. (Figure 7B, Appendix A). The regulatory differences of key genes in various metabolic pathways are shown in (Appendix A). Generally speaking, most of these key genes were up-regulated, such as citrate synthase (*CS*), isocitrate dehydrogenase (*IDH1*) and 2-oxoglutarate dehydrogenase (*OGDH*) in glycolysis, acetyl CoA synthase (*ACSS*) and pyruvate carboxylase (*PC*) in pyruvate metabolism, malonyl CoA acyl carrier protein acyltransferase (*MCAT*) and long-chain acyl CoA synthetase (*acsA*) in fatty acid biosynthesis, and most genes in amino acid metabolism.

### 3.6. RT-qPCR of the Key Genes Related to MPs Biosynthesis

The biosynthesis of MPs is mainly related to the FAS pathway and the PKS pathway [5,6]. RT-qPCR was used to verify the expression levels of key secondary metabolic genes involved in MPs biosynthesis. As shown in Figure 8, the relative expression levels of *MpPKS5*, *MpFasA2*, *MpFasB2*, *mppA*, *mppB*, *mppC*, *mppD*, *mppE* and *mppR1* regulated by T11 were significantly up-regulated compared with the Con group (Figure 8), which was basically consistent with the results of transcriptome sequencing (Appendix A). Meanwhile, the expression level of *mppR2* in the T11 group was significantly lower than that in the Con group, which was slightly different from the results of transcriptome sequencing. Among these key genes, *MpFasA2* and *MpFasB2* are responsible for producing the side chain fatty acyl moiety of MPs [5], and the genes *MpPKS5* and *mppD* are involved in the polyketide chromophore biosynthesis of pigments [31]. A previous report showed that knock-out of *MpPKS5* will lead to a very low MPs yield [32]. Moreover, *mppB* encodes fatty acyltransferase, which was involved with the synthesis of pigment precursor of hexone chromophore and β-keto acid. Furthermore, *mppC* encodes oxidoreductase, which is positively correlated with yellow pigment metabolism. These results can be confirmed in the NR database and KEGG metabolic pathway. The up-regulation of these genes indicated that T11 could promote the secondary metabolic pathway of MPs synthesis in *M. purpureus* M3 and ultimately act to increase MPs production.

### 3.7. Metabolomic Analysis Reveal the Regulatory Effects of T11 on MPs Biosynthesis

The metabolites of mycelium were detected on the 14th day of fermentation by GC-TOF-MS. The results of mycelium metabolites analyzed by PCA and PLS-DA showed a significant difference of the metabolites between the Con and T11 groups (Figure 9A,B). Correspondingly, the hierarchical cluster analysis illustrated that the metabolites between the Con and T11 samples were also separated obviously (Figure 9C). As shown in Figure 9D, the metabolites with log_2_(FC) > 1.0 and *p* value < 0.05 far from the center in the load diagram were considered as obviously altered mycelium metabolites (potential biomarkers) in group T11, which were the responsible for the difference between the Con group and the T11 group. A total of 115 potential biomarkers in mycelium were successfully identified (Figure 9F). Compared with the Con group, 81 and 34 metabolites were significantly up-regulated and down-regulated in the T11 group, respectively. Specifically, up-regulated metabolites mainly included citramalic acid, capric acid, 3-phosphoglycerate, caprylic acid, fructose 6-phosphate, 6-phosphogluconic acid, oleic acid, asparagine, linoleic acid, glutathione, arachidic acid, homogentisic acid, aspartic acid, fumaric acid, glycerol, L-glutamic acid and tartaric acid, etc. On the contrary, the down-regulated metabolites mainly included pyruvic acid, lactic acid, isoleucine, valine, proline, alanine, glycine and succinic acid, etc. To further investigate the metabolic changes in response to T11 intervention, Mbrole2.0 was used to analyze the main enrichment pathway of DMs to clarify the main metabolic pathways affected (Figure 9E). The result revealed that a total of 45 metabolic pathways were enriched. Among these, the DMs were mainly enriched in alanine, aspartate and glutamic acid metabolism, starch and sucrose metabolism, glycine, serine and threonine metabolism, arginine and proline metabolism, and glutathione metabolism, etc., indicating that these pathways were significantly affected after T11 intervention.

## 4. Discussion

The secondary metabolites of *Monascus* spp. can be affected by many factors, such as the composition of the medium and the fermentation environment during the fermentation process [33,34,35]. In recent decades, many scholars have devoted themselves to increasing MPs production by improving fermentation conditions and medium composition. Previous studies have reported that high glucose stress could promote the secretion of intracellular pigment and transmembrane transformation, and the total production of extracellular and intracellular yellow pigment increased by 94.86% and 26.31%, respectively [35]. Inorganic nitrogen sources have also been proved to improve mycelial morphology and promote the biosynthesis of MPs [23]. Soluble starch and glycerol could enhance the production of MPs, confirmed by comparative proteomic and transcriptional analyses [7]. Some flavonoids such as genistein have been proved to promote the production of MPs in previous studies [36]. However, the price and difficulty of obtaining standard compounds may be an obstacle for industrial application, which will increase the cost of MPs industrial production. Therefore, in this study, the economical tea extract was used as the regulatory factor to study its effect and regulatory mechanism on MPs biosynthesis.

The process of MPs biosynthesis is usually that acetyl-CoA and malonyl-CoA to generate hexanone chromophore under the catalysis of PKS. Then, the hexanone chromophore and the medium-chain fatty acids produced by the fatty acid synthesis pathway undergo a series of processes (such as transesterification reaction, reduction reaction and ammonia reaction) to generate different types of MPs [5,6]. The biosynthesis of MPs involves the metabolism of several substances, such as acetyl-CoA, malonyl-CoA and medium-chain fatty acids (such as caprylic acid and capric acid, etc.), which are precursors of pigment synthesis and have direct effects on pigment biosynthesis. Transcriptome sequencing and metabolome results annotated by the KEGG database showed that T11 intervention significantly affected *M. purpureus* M3 metabolism, mainly involving the tricarboxylic acid (TCA) cycle, glycolysis, amino acid metabolism, pyruvate metabolism and fatty acid synthesis pathways, accompanied by significant changes in a large number of related genes and metabolites.

In the glycolysis/gluconeogenesis pathway, the *GALM*, *TPI*, *ENO* and *pckA* genes were significantly up-regulated in the T11 group, while the *HK* and *gpmI* genes were significantly down-regulated. The *GALM* and *TPI* genes encode galactose gyrase and triosephosphate isomerase, respectively. *ENO* encodes enolase, which assists in the conversion of 2-phosphoglyceric acid to phosphoenolpyruvate. The *p**ckA* gene encodes phosphoenolpyruvate carboxylase, which catalyzes oxaloacetate to phosphoenolpyruvate, and is the rate-limiting enzyme of gluconeogenesis. Up-regulation of these genes may mean that the glycolysis/gluconeogenesis pathway is more active under the intervention of T11. The *HK* gene encodes hexokinase, which can catalyze glucose to produce glucose-6-phosphate. It is a key gene in glycolysis and participates in the first reaction of glycolysis [37]. The down-regulation of the *HK* gene does not necessarily mean that glycolysis is inhibited; it may be the product inhibition caused by excessive consumption of glucose and accumulation of high concentrations of glucose 6-phosphate in glycolysis [38]. Correspondingly, the down-regulation of glucose concentration in metabolomics results can also indirectly prove this point. Moreover, previous studies have shown that the overexpression of the *pckA* gene and *TPI* gene can enhance the pyruvate anabolic pathway, promote the metabolic capacity of strain and increase the production of pyruvate [39].

In pyruvate metabolism, the relative expression levels of the *DLD*, *PC*, *ACOT12*, *ALDH* and *ACSS* genes in the T11 group were significantly up-regulated compared with the Con group. These genes encode dihydrolipoyl dehydrogenase, pyruvate carboxylase, acetyl-CoA hydrolase, acetaldehyde dehydrogenase and acetyl-CoA synthase, respectively, indicating that T11 promoted the pyruvate metabolism of *M. purpureus* M3 in liquid fermentation. Pyruvate can be converted into oxaloacetic acid by pyruvate carboxylase to participate in the TCA cycle, which may be the reason for the down-regulation of pyruvate concentration in the T11 group. Moreover, pyruvate can also be reduced to lactate or oxidized to acetyl-CoA. However, the relative expression of the *LDH* gene (which encodes lactate dehydrogenase) in the T11 group was significantly down-regulated, illustrating that the intervention of T11 can reduce the consumption of pyruvate caused by the production of lactate. It was worth noting that the *CS*, *ACO*, *IDH1*, *OGDH*, *fumC* and *MDH2* genes were significantly up-regulated by T11 intervention. These genes encode key enzymes of the TCA cycle, and their up-regulation indicates that T11 promotes the TCA cycle and increases the primary metabolic capacity of *M. purpureus* M3 to produce more energy substances. This result is consistent with the effect of T11 on pyruvate metabolism. In addition, compared with the control group, the concentrations of citric acid and fumarate were up-regulated in T11, whereas succinate was down-regulated. Citrate and fumarate were found to be up-regulated in the T11 group, which was consistent with the trend in related genes in the TCA cycle. The increase in fumarate may be responsible for the down-regulation of succinate at T11, since succinate can be converted to fumarate.

In the fatty acid synthesis pathway, the expression of the malonyl-CoA-acyl carrier protein transacylase (*MCAT*) and long chain acyl-CoA synthetase (*acsA*) was significantly up-regulated. Correspondingly, fatty acids such as caprylic acid and capric acid were also found to be up-regulated in metabolomics results. Caprylic acid and capric acid are precursors of MPs synthesis, which can promote the biosynthesis of MPs [23]. These results indicated that the supplementation of T11 can promote the fatty acid metabolism pathway of *M. purpureus* M3, resulting in the increase in fatty acids to provide more precursors for the synthesis of MPs.

Compared with the Con group, the main affected amino acid metabolism of the T11 group were alanine, tyrosine, glutamate and leucine metabolism, etc. Alanine can be converted to pyruvate under the action of alanine-glyoxylate aminotransferase (*AGXT*) and alanine aminotransferase (*ALT*), and then enters pyruvate metabolism to generate more acetyl-CoA. Corresponding to the transcriptome results, both alanine and pyruvate were down-regulated after T11 intervention in the metabolome, which may suggest that they were used to produce more acetyl-CoA or enter the TCA cycle for energy production. Tyrosine, tryptophan and phenylalanine are aromatic amino acids that can be converted into several biological compounds (such as 5-hydroxy tryptamine, heteroauxin, dopamine, melanin, etc.) through a series of oxidation reactions to participate in the growth process and substance metabolism. As for glutamate metabolism, the enhancement of transcription on 4-aminobutyrate transaminase (*ABAT*), succinate semialdehyde dehydrogenase (*ALDH5A1*) and glutarate-semialdehyde dehydrogenase (*gabD*) were significant, and the final product, succinate, could enter the TCA cycle. Finally, in valine, leucine and isoleucine metabolism, these branched-chain amino acids can be degraded by related enzymes encoded by DEGs (*BCKDHB*, *ACADM*, *E6.4.1.4**b*, *Hmgcll1* and *OXCT*) to generate acetyl-CoA and methyl-CoA, which can be further catalyzed by PKS to form type II polyketide compounds, the precursors of MPs synthesis [40]. Interestingly, isoleucine and valine also showed down-regulation in the metabolomic results compared with controls, which could be used to explain the transcriptomic results. In addition, glutathione, a thiol-containing tripeptide composed of glutamate, cysteine and glycine, was also found to be up-regulated in the T11 group, mainly in the reduced form of glutathione (GSH). GSH can also participate in the TCA cycle and glucose metabolism, activate a variety of enzymes and promote the metabolism of sugar, fat and protein. The down-regulation of GST expression may be the reason for the high accumulation of GSH in the T11 group. In summary, in the above amino acid metabolism, the general trend is to produce more substances related to the TCA cycle, including pyruvate, fumaric acid, succinic acid and acetyl-CoA, thus further promoting the activity of the TCA cycle.

## 5. Conclusions

In this study, we explored the effect of 15% ethanol extract of pu-erh tea (T11) intervention on *M. purpureus* M3, based on transcriptomics combined with metabolomics, and provided a basis for further understanding the regulation of MPs synthesis. After T11 intervention, the changes of gene transcription level and metabolites in *M. purpureus* M3 were basically consistent, and mainly concentrated in the primary metabolic network. RT-qPCR validation results showed that the expression of key genes in the secondary metabolism of MPs was also up-regulated by T11, thereby enhancing the production of MPs. The low economic value of tea extract may make it suitable for industrial production. However, since tea extract is a mixture rich in polyphenols, it is difficult to explain exactly which compound plays the regulatory role, which is also a limitation of this work for exploring the deep mechanism. In the future, we may further study the effect of specific tea polyphenols on MPs synthesis, so as to further understand the molecular mechanism of tea polyphenols on the synthesis and metabolism of MPs. This study is expected to lay the foundation for a more systematic and comprehensive study of the molecular mechanism of different substances affecting the metabolism of *Monascus* spp.

## Figures and Tables

**Figure 1 foods-11-03159-f001:**
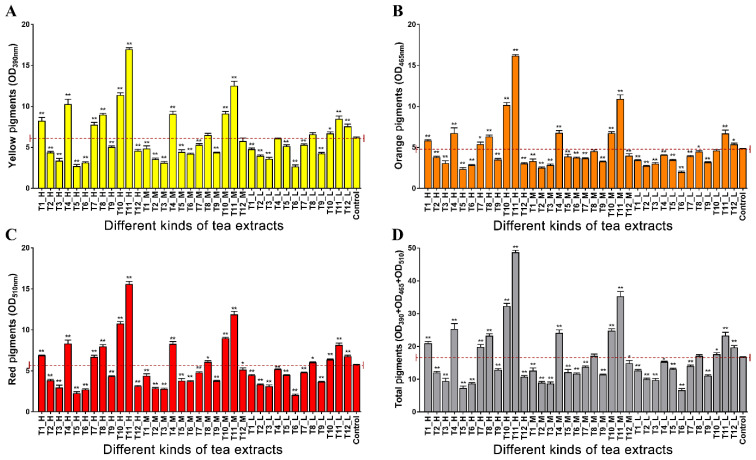
Effect of tea extracts on MPs production of *M. purpureus* M3. (**A**–**D**) represent the color values of yellow pigments, orange pigments, red pigments and total pigments, respectively. Color values were used to measure the production of pigments among different groups. T1–T3, T4–T6, T7–T9 and T10–T12 are extracts of green tea, black tea, white tea and pu-erh tea in water, 15% ethanol and 75% ethanol, respectively. H, M and L represent the total polyphenol concentrations of 250 μg/mL, 125 μg/mL and 62.5 μg/mL in tea extracts, respectively. The red dashed line represents the average color value of the control group. Note: * *p* < 0.05 and ** *p* < 0.01 versus the Con group.

**Figure 2 foods-11-03159-f002:**
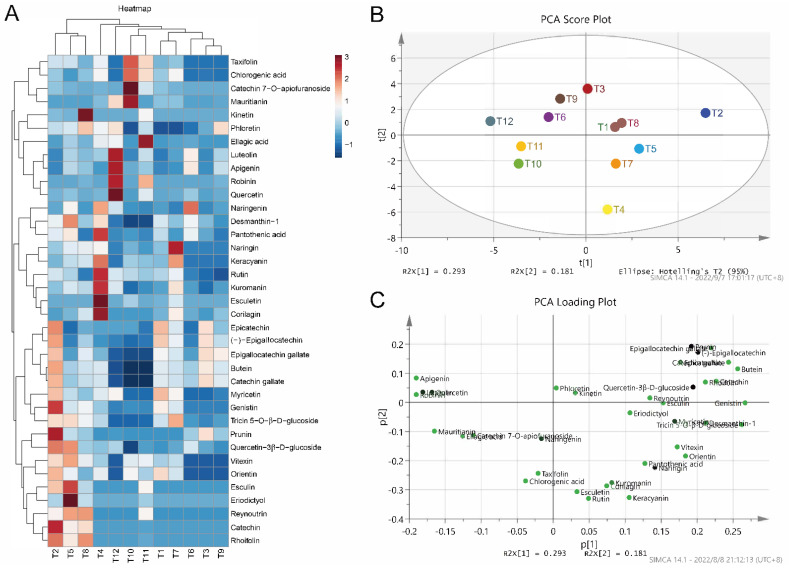
Composition analysis of 12 tea extracts. T1–T3, T4–T6, T7–T9 and T10–T12 are extracts of green tea, black tea, white tea and pu-erh tea in water, 15% ethanol and 75% ethanol, respectively. Heatmap of the components (**A**), PCA score plot (**B**) and PCA loading plot (**C**).

**Figure 3 foods-11-03159-f003:**
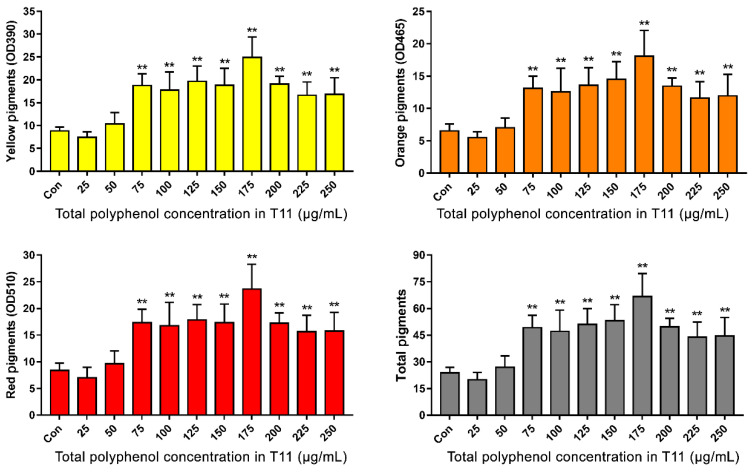
Effects of T11 (pu-erh tea extracted with 15% ethanol) at different total polyphenols concentrations on MPs production. Note: ** *p* < 0.01 versus the Con group.

**Figure 4 foods-11-03159-f004:**
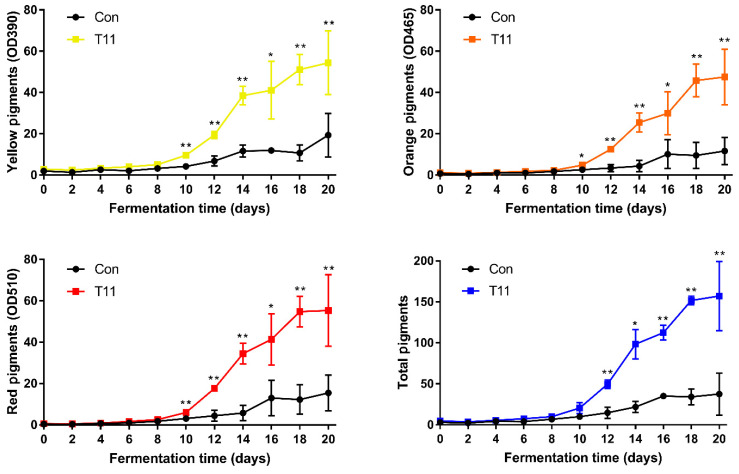
Effect of fermentation time on MPs production under optimal supplemental concentration of T11 (pu-erh tea extracted with 15% ethanol). Note: * *p* < 0.05 and ** *p* < 0.01 versus the Con group.

**Figure 5 foods-11-03159-f005:**
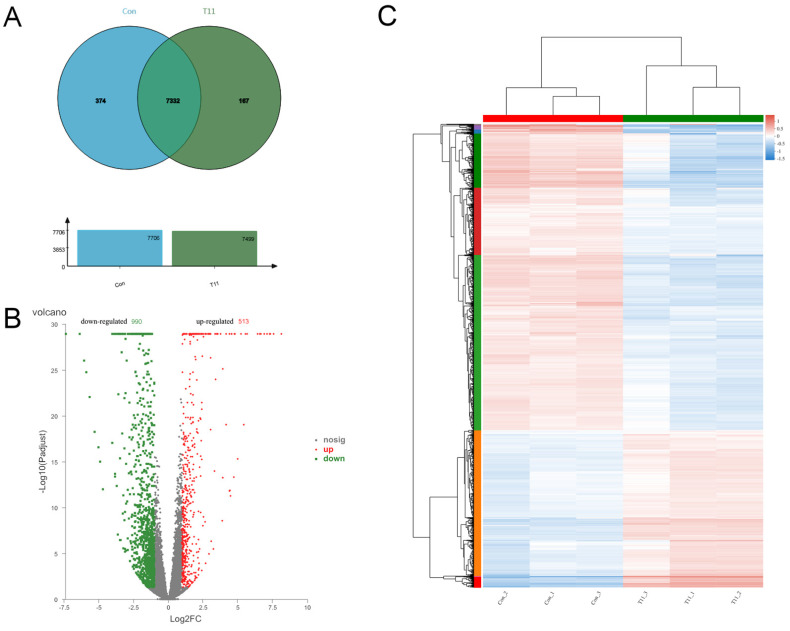
The Venn diagram, volcano plots and cluster analysis of DEGs between Con and T11 (pu-erh tea extracted with 15% ethanol) groups. (**A**): the Venn diagram of gene expression (**B**): the volcano diagram of DEGs. (**C**): the clustering diagram of DEGs.

**Figure 6 foods-11-03159-f006:**
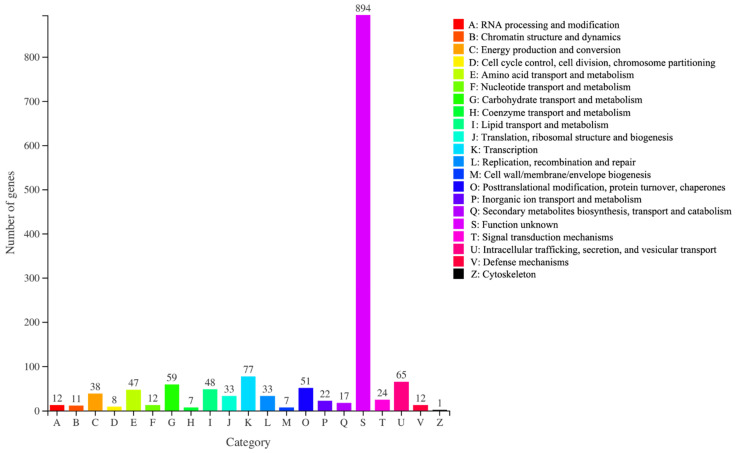
COG functional classification of DEGs in “Con vs T11”.

**Figure 7 foods-11-03159-f007:**
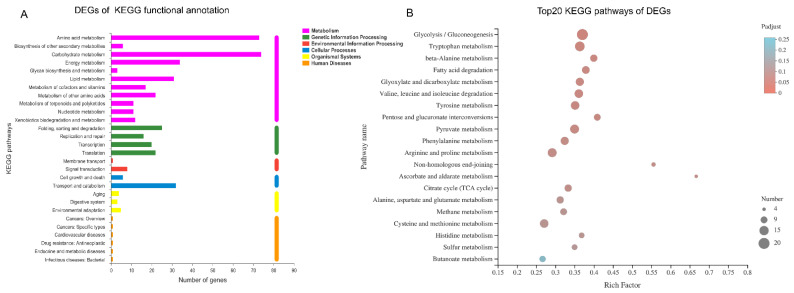
(**A**): KEGG functional classification of DEGs in “Con vs T11”. (**B**): KEGG pathway enrichment of in “Con vs T11”. The rich factor represents the number of DEGs relative to the percentage of all annotated genes involved in the pathway.

**Figure 8 foods-11-03159-f008:**
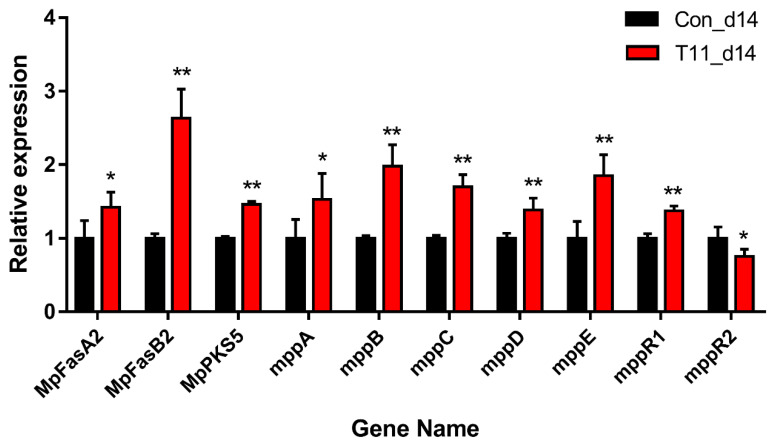
Relative expression levels of MPs-biosynthesis-related genes in Con group and T11 group on 14th day based on RT-qPCR. The transcriptional levels were normalized to β-actin gene. Note: * *p* < 0.05 and ** *p* < 0.01 versus the Con group.

**Figure 9 foods-11-03159-f009:**
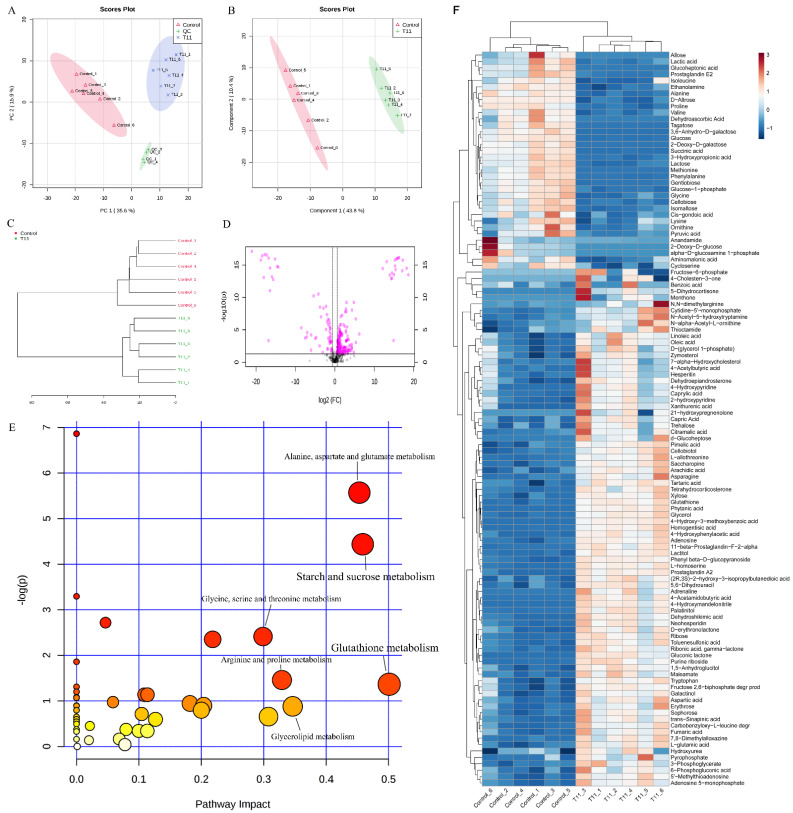
Mycelium metabolomic profiling by GC-TOF-MS. (**A**) PCA score plot of the Con and T11 groups; (**B**) PLS-DA score plot of the Con and T11 groups; (**C**) hierarchical cluster map of the Con and T11 groups; (**D**) the volcano plots of the Con and T11 groups; (**E**) the metabolic pathway impact prediction between the Con and T11 groups in mycelium based on the KEGG online database. The −ln(p) values from the pathway enrichment analysis are indicated on the vertical axis, and the impact values are indicated on the horizontal axis; the smaller the *p*-value, the redder the circle color, and the larger the influence value, the larger the circle; (**F**) heatmap of relative abundance of significant different metabolites (log_2_(FC) > 1.0, and *p* < 0.05) in mycelium on the 14th day of fermentation in the Con and T11 groups.

**Table 1 foods-11-03159-t001:** Different tea extracts and their corresponding experimental numbers.

Extraction Solvent	Green Tea	Black Tea	White Tea	Pu-Erh Tea
Water	T1	T4	T7	T10
15% ethanol	T2	T5	T8	T11
75% ethanol	T3	T6	T9	T12

## Data Availability

The data presented in this study are available on request from the corresponding author.

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
