# Peer review of "Comparative Transcriptomic and Metabolomic Analyses Reveal the Regulatory Effect and Mechanism of Tea Extracts on the Biosynthesis of Monascus Pigments"

_foods, 2022, doi:10.3390/foods11203159_

Round 1
Reviewer 1 Report
The authors reported the regulatory effects of different tea extracts on MPs biosynthesis and revealed the mechanism of the regulatory effects by comparative transcriptomic and metabonomic analyses. This results demonstrated that the ethanol extract of Pu’er tea enhanced MPs production by affecting the primary metabolic pathway, providing sufficient energy and more biosynthetic precursors for secondary metabolism. This work will benefit comprehensive understanding of the regulation of tea polyphenols on MPs biosynthesis in Monascus spp.
However, the text should be revised to add clarity, bolster the scientific interpretation and polish the language (sentence structure, word choice, grammar and syntax), in addition to removing speculative statements.
1. The abstract should be refined to present the most important results and conclusion.
2. Line 16: What is the meaning of the phrase “high nutrition”?
3. Line 40-44: The introduction of Hongqu is not related to this study and should be deleted.
4. Line 47: “On the contrary” is not correctly used.
5. Line 79-95, 99-103: The common knowledge of omics techniques should not be presented in details in the section of introduction. Instead, the research progresses on MPs biosynthesis using omics methods should be reviewed.
6. Line 118: The centrifugation speed should be expressed by “g” instead of rpm.
7. Line 130: enzyme-labeled instrument?
8. Line 143: 10d?
9. Line 163: prepared as this concentration?
10. Line 177: What dose Q20 and Q30 mean?
11. Line 238-239: there were significant difference ?
12. Line 258-259: “Simply put, the source and treatment of tea extracts will affect its compound composition.” Common sense should not be presented in the results.
13. Line 276-277: The authors found that T11 were high concentrations of ellagic acid, taxifolin and chlorogenic acid, which should be the potential compounds for MPs production promotion. Did the authors use standard samples of these compounds to test their effects on MPs production?
14. Line 289-290: “It was worth noting that the production of MPs increased most significantly at the concentration of T11 was 175 μg/mL”. Wrong grammar.
15. Fig 4: different fermentation stages?
16. Line 338-339: The description should be moved to the methods section.
17. Line 358-359: “The further analysis…need to” ?
18. Line Fig 6 &7: “Con cs T11”?
19. Line 400-404: The brief summary repeated the aforesaid results, so it should be deleted.
20. Discussion Section: Some contents should be moved to the results section, such as the Transcriptome sequencing and metabolome results analysis.
21. Line 465-471: The contents should be presented in the introduction section.
22. Line 582-583: “so as to further understand the molecular mechanism of tea polyphenols on the synthesis and metabolism of mucopolysaccharides”. Is mucopolysaccharide related to this study?
Author Response
Thank you very much for your comments, which are very helpful to improve the quality of our manuscript.
Review expert 1
The authors reported the regulatory effects of different tea extracts on MPs biosynthesis and revealed the mechanism of the regulatory effects by comparative transcriptomic and metabonomic analyses. This results demonstrated that the ethanol extract of Pu’er tea enhanced MPs production by affecting the primary metabolic pathway, providing sufficient energy and more biosynthetic precursors for secondary metabolism. This work will benefit comprehensive understanding of the regulation of tea polyphenols on MPs biosynthesis in Monascus spp.
However, the text should be revised to add clarity, bolster the scientific interpretation and polish the language (sentence structure, word choice, grammar and syntax), in addition to removing speculative statements.
Q1. The abstract should be refined to present the most important results and conclusion.
A1: Thanks for your constructive comment. We have revised the abstract based on your comment and presented the more important results. In addition, we have supplemented a description of the novelty of this study in the abstract.
Q2. Line 16: What is the meaning of the phrase “high nutrition”?
A2: Thanks for your rigorous comment to help us point out the misnomer here. We have corrected it.
- Line 40-44: The introduction of Hongqu is not related to this study and should be deleted.
A3: Thanks for your comment. We have revised the introduction accordingly.
Q4. Line 47: “On the contrary” is not correctly used.
A4: Thanks for your comment. We have deleted the inappropriate expressions here.
Q5. Line 79-95, 99-103: The common knowledge of omics techniques should not be presented in details in the section of introduction. Instead, the research progresses on MPs biosynthesis using omics methods should be reviewed.
A5: Thanks for your comment. We think it's very helpful to us. In this section, we have simplified the common knowledge of omics techniques and focused on reviewing the progress of previous studies applied omics techniques in MPs.
Q6. Line 118: The centrifugation speed should be expressed by “g” instead of rpm.
A6: Thanks for your comment. We have converted “rpm” into “g” to describe the centrifugal state based on the rotor radius.
Q7. Line 130: enzyme-labeled instrument?
A7: Thanks for your comment. Perhaps our expression here is not standard enough, we have changed it to “Multi-Mode Detection Platform”.
Q8. Line 143: 10d?
A8: Thanks for your comment. Perhaps our expression here is not standard enough, we have changed it to “10 days”.
Q9. Line 163: prepared as this concentration?
A9: Thanks for your serious suggestion, which can improve the quality of our manuscript. Maybe our expression here is not clear enough,we have modified accordingly as follows: “After the optimal supplemental concentration of tea extract was selected, the optimal concentration of tea extract was used as the supplement of culture medium to study the effect of fermentation time on MPs production”.
Q10. Line 177: What does Q20 and Q30 mean?
A10: Thanks for your comment. I'm sorry we didn't explain Q20 and Q30 here. The higher the percentage value of quality values "Q20 and Q30", the lower the probability of base error detection. Generally speaking, if the Q30 value of sequencing data is greater than 85%, the sequencing date is considered to be reliable. We believe that these implications are common knowledge in the application of sequencing technology and are not the focus of this study. And it is easy to retrieve relevant knowledge in the browser, so the mean of Q20 and Q30 was not specified in the method.
Q11. Line 238-239: there were significant difference ?
A11: Thanks for your comment. I am sorry that the expression here may not be appropriate, because there was no statistical analysis between the different tea extract treatment groups. So we have changed "significant" to "obvious" accordingly.
Q12. Line 258-259: “Simply put, the source and treatment of tea extracts will affect its compound composition.” Common sense should not be presented in the results.
A12: Thanks for your comment. We have deleted the redundant part here.
Q13. Line 276-277: The authors found that T11 were high concentrations of ellagic acid, taxifolin and chlorogenic acid, which should be the potential compounds for MPs production promotion. Did the authors use standard samples of these compounds to test their effects on MPs production?
A13: Thanks for your constructive comment. We are sorry that we don’t use standard compounds to test their effects on MPs production in this work. In this study, we aimed to investigate the effect of tea extract on MPs production and explore the key regulatory pathways. In addition, we also preliminarily analyzed the potential promoters of MPs synthesis in tea extract. And the confirmatory experiments on the effects of these potential promoters on MPs synthesis are the direction of our next work.
Q14. Line 289-290: “It was worth noting that the production of MPs increased most significantly at the concentration of T11 was 175 μg/mL”. Wrong grammar.
A14: Thanks for your constructive comments. I'm sorry for the grammatical error here. In this regard, we specially invited an English major friend to help us revise it as follow: Notably, the most significant increase in MPs production was observed at a T11 concentration of 175 μg/mL.
Q15. Fig 4: different fermentation stages?
A15: Thanks for your careful review and help us find inappropriate statements. We have changed it to "Effect of fermentation time on MPs production under optimal supplemental concentration of T11".
Q16. Line 338-339: The description should be moved to the methods section.
A16: Thanks for your comment. We have moved the description here to the methods section according to your comments.
Q17. Line 358-359: “The further analysis…need to” ?
A17: Thanks for your comment to help us point out inappropriate sentences here. After checking the original text, we also think that this sentence should not be placed here. Therefore, we deleted this redundant sentence here.
Q18. Line Fig 6 &7: “Con cs T11”?
A18: Thanks for your careful advice. we are sorry for the error here and have corrected the manuscript accordingly.
Q19. Line 400-404: The brief summary repeated the aforesaid results, so it should be deleted.
A19: Thanks for your comments. We have deleted the duplicate instructions accordingly. In addition, we have supplemented the functional information of these key genes for pigment synthesis to illustrate the necessity of selecting these key genes in this section.
Q20. Discussion Section: Some contents should be moved to the results section, such as the Transcriptome sequencing and metabolome results analysis.
A20: We agree with your constructive suggestion, and try our best to transfer some discussions to the results section, including the function introduction of key genes in MPs synthesis, etc. However, because the discussion contents of the two omics are the main part of the discussion, we can't transfer all of them to the results, which will lead to the results being too long and not clear and concise enough.
Q21. Line 465-471: The contents should be presented in the introduction section.
A21: Thanks for your comment. We have transferred the appropriate content from here to the introduction in accordance with your comments.
Q22. Line 582-583: “so as to further understand the molecular mechanism of tea polyphenols on the synthesis and metabolism of mucopolysaccharides”. Is mucopolysaccharide related to this study?
A22: Thans for your careful comment. The “Mucopolysaccharide” is not related to this study. I am very sorry for this clerical error and have revised the manuscript accordingly.

Reviewer 2 Report
The paper “Comparative transcriptomic and metabonomic analyses reveal the regulatory effect and mechanism of tea extracts on the biosynthesis of Monascus pigments” contributes to the growth of literature for research on bioactive food.
Before the manuscript acceptation for publication in “Foods” the following items should be revised:
Typing errors must be corrected, e.g. Eeffect
The description of the aim of the Review is not specific.
Figures
I suggest expanding the titles, e.g. Figure 1. "kinds of tea extracts" - the title should contain information about the type of extraction and the type of tea)
Figure 2
the graph is not very clear, especially B-D
Figure 3
I suggest expanding the titles - changing the symbol T11 to the type of tea and extraction - similarly to Tables 5 – 9.
Figure 4 The Title Eeffect of different fermentation stages on the yield of MPs.
“Eeffect” to “Effect” and “different fermentation stages” what fermentation stages
Discussion
“The up-regulation of these genes indicated that T11 could promote the secondary metabolic pathway of MPs synthesis in M. purpureus M3 and ultimately act to increase MPs production.” - Do authors know why - based on literature?
Conclusions
The conclusions are similar to the abstract.
I suggest changing the symbol of the tea and the extract with the name, e.g. T11 to 15% ethanol of Pu-erh tea
What are the limitations of this method?
The authors should add the summary conclusion - the positive or negative effects of the research, what recommendations are for the future of food and this research, especially to "a solid foundation for the industrial-scale production of MPs. - Line 108"
Author Response
Thank you very much for your comments, which are very helpful to improve the quality of our manuscript.
Review expert 2
The paper “Comparative transcriptomic and metabonomic analyses reveal the regulatory effect and mechanism of tea extracts on the biosynthesis of Monascus pigments” contributes to the growth of literature for research on bioactive food.
Before the manuscript acceptation for publication in “Foods” the following items should be revised:
Q1:Typing errors must be corrected, e.g. Eeffect
A1: Oh, we are very sorry for the slip of pen here. Thank you very much for pointing it out.
Q2:The description of the aim of the Review is not specific.
A2: Thanks for your excellent comment. We have supplemented a more specific description of the aim of the review in introduction section based on your comments. In addition, we also revised the omics technology review section, deleting the tedious omics basic knowledge and focusing on the application and progress of omics technology in MPs research.
Figures
Q3:I suggest expanding the titles, e.g. Figure 1. "kinds of tea extracts" - the title should contain information about the type of extraction and the type of tea
A3: Thanks for your comment. We have made more specific additions to expand the titles of pictures and revised the manuscript accordingly.
Q4:Figure 2 the graph is not very clear, especially B-D
A4: Thanks for your comment. We are sorry for the poor reading experience caused by the unclear picture. To improve this situation, we adjusted Figure 2 accordingly. Firstly, Hierarchical Cluster plot is transferred to the supplementary file to give larger display space to Figure 2B and Figure 2C. Second, we have enlarged the fonts in Figures 2A and 2C for clearer viewing.
Q5:Figure 3
I suggest expanding the titles - changing the symbol T11 to the type of tea and extraction - similarly to Tables 5 – 9.
A5: Thanks for your excellent comment. We have tried our best to expand the title according to your comment.
Q6:Figure 4 The Title Eeffect of different fermentation stages on the yield of MPs.
“Eeffect” to “Effect” and “different fermentation stages” what fermentation stages
A6: Thanks for your careful review and help us find inappropriate statements and clerical error. We have changed it to "Effect of fermentation time on MPs production under optimal supplemental concentration of T11".
Discussion
Q7:“The up-regulation of these genes indicated that T11 could promote the secondary metabolic pathway of MPs synthesis in M. purpureus M3 and ultimately act to increase MPs production.” - Do authors know why - based on literature?
A7: Thanks for your comment. We have transferred the imoformations of key genes for MPs synthesis into the results in 3.6 section. In addition, we have mentioned and cited relevant references in the introduction, so we do not repeat references here. In fact, we mainly identified these key genes based on two review articles published by Chen et al., and applied them in the manuscript. In addition, some corresponding studies of other scholars were supplemented in the manuscript.
References:
Chen, W.P.; Chen, R.F.; Liu, Q.P.; et al. Orange, red, yellow: Biosynthesis of azaphilone pigments in Monascus fungi. Chem Sci 2017; 8(7): 4917–4925.
Chen, W.P.; Feng, Y.L.; István.; et al. Nature and nurture: confluence of pathway determinism with metabolic and chemical serendipity diversifies Monascus azaphilone pigments. Nat Prod Rep 2019; 36(4): 561-572.
Conclusions
Q8:The conclusions are similar to the abstract.
I suggest changing the symbol of the tea and the extract with the name, e.g. T11 to 15% ethanol of Pu-erh tea
A8: Thanks for your wonderful comment. We have mentioned in the conclusion that T11 is a tea extract extracted with 15% ethanol according to your comments. In addition, we also mention the limitations of this work and future research directions in the conclusion.
Q9: What are the limitations of this method?
The authors should add the summary conclusion - the positive or negative effects of the research, what recommendations are for the future of food and this research, especially to "a solid foundation for the industrial-scale production of MPs. - Line 108"
A9: Thanks for your excellent comment. In this study, supplementation of tea extract was used to enhance the yield of monascus pigment, which is undoubtedly cheaper than the supplementation of standard compounds in terms of economic price. However, this study did not investigate which particular compounds in tea extracts were responsible for increasing MPs production. Therefore, the lack of confirmatory experiments on specific standard compounds is the limitation of this research method, which is also a direction of our future research. However, because of the convenience and low economic value of tea extract, it may be more easily used in industrial production. Maybe the word "solid foundation" here is not quite appropriate, we have changed it to "potential application foundation".

Reviewer 3 Report
In the manuscript, the authors studied screening of tea extract having effect on Monascus pigments and analysis for the promoting effect. The data were well performed and interpreted. I have several questions and suggestions before the paper is decided to publish on this journal.
1. Line 24: What “NR” means? Please describe the formal name.
2. Line 50: polyketene. Is it precise? Polyketide? And, here, the authors set the abbreviation PKS, so please use PKS below in line 391, 466, 542, and 559. Please revise on FAS as same.
3. Line 243, 244, and caption of Figure 1 and more: The authors describe two styles of p value (p or P). And, Should “p value” be written in italic? Please unify and revise them.
4. Line 267: catechin 7-o-apiofuranoside => Should “o” be written in italic? And, it is written as capital letter in the Figure 2A.
5. Line 275-280: “Compared with other tea extracts with poor or negative effects on MPs production, ... to promote MPs production 277 (Fig. 2A). Of course, ... in other 279 tea extracts.” I agree that ellagic acid is a promoting factor in T11. On the other hand, I guess that catechin 7-o-apiofuranoside and mauritianin may be promoting factors. If they are not, were the reasons explained and illustrated in the Figure 2BCD? They are in same cluster with ellagic acid in the Figure 2C. As same, how about the potential of taxifolin and chlorogenic acid? They are separated from the ellagic acid cluster.
6. Line 466: Please insert a space between polyketide and synthase.
7. Line 577: polyketo => polyketide
Author Response
Thank you very much for your comments, which are very helpful to improve the quality of our manuscript. The specific reply is shown in the attachment. We have also uploaded a reply on the website for your review.
Review expert 3
In the manuscript, the authors studied screening of tea extract having effect on Monascus pigments and analysis for the promoting effect. The data were well performed and interpreted. I have several questions and suggestions before the paper is decided to publish on this journal.
Q1. Line 24: What “NR” means? Please describe the formal name.
A1: Thanks for your comment. After careful consideration of your suggestion, we think that the completion of the database name may cause the summary to be too verbose. Therefore, KEGG and NR were deleted from the abstract.
Q2. Line 50: polyketene. Is it precise? Polyketide? And, here, the authors set the abbreviation PKS, so please use PKS below in line 391, 466, 542, and 559. Please revise on FAS as same.
A2: Thanks for your careful comment. I'm sorry that our professional terms are not unified here, and we have modified these terms accordingly.
Q3. Line 243, 244, and caption of Figure 1 and more: The authors describe two styles of p value (p or P). And, Should “p value” be written in italic? Please unify and revise them.
A3: Thanks for your comment, which helped us point out the shortcomings of our manuscript. We have uniformly capitalized the "P" and written it in italics.
Q4. Line 267: catechin 7-o-apiofuranoside => Should “o” be written in italic? And, it is written as capital letter in the Figure 2A.
A4: Thanks for your careful comment. I'm sorry we have a clerical discrepancy here. After searching the compound information, we confirmed that the “O” in Catechin 7-O-apiofuranoside should be uppercase and non-italic, which we have corrected and unified in the manuscript.
Q5. Line 275-280: “Compared with other tea extracts with poor or negative effects on MPs production, ... to promote MPs production 277 (Fig. 2A). Of course, ... in other 279 tea extracts.” I agree that ellagic acid is a promoting factor in T11. On the other hand, I guess that catechin 7-o-apiofuranoside and mauritianin may be promoting factors. If they are not, were the reasons explained and illustrated in the Figure 2BCD? They are in same cluster with ellagic acid in the Figure 2C. As same, how about the potential of taxifolin and chlorogenic acid? They are separated from the ellagic acid cluster.
A5: Thanks for this constructive comment. This issue is also of concern to us. Because only by substance concentration, ellagic acid in T11 is the most obvious higher than other tea extracts, and is the characteristic substance in T11. In addition, I also agree with you that catechin 7-o-apiofuranoside and mauritianin are also facilitators, as they are in a similar position to ellagic acid in the loading plot. However, according to the heat map (Figure 2A), the concentrations of these two substances in T11 were obviously lower than those in tea extract T10, so they may be potential factors in 10 to promote the synthesis of MPs. Therefore, although they are in the same quadrant as ellagic acid in the loading plot, the concentration of these two substances in T11 is relatively low, so we did not mention these two substances as promoting factors in tea extract T11 in the paper. As for taxifolin and chlorogenic acid, we also consider them as potential promoters, as the concentrations of both substances were relatively high in the two groups of tea extracts (T1 and T11) that were most effective in promoting MPs production (fig. 2A). The effect of these specific standard compounds on MPs will be the direction of our subsequent research.
Q6. Line 466: Please insert a space between polyketide and synthase.
A6: Thanks for your careful comment. We have revised accordingly in the manuscript.
Q7. Line 577: polyketo => polyketide
A7: Thanks for your comment, which helped us point out the error here. We have revised accordingly in the manuscript.

Reviewer 4 Report
Manuscript proposed by Li and co-workers (foods-1902255) entitled “Comparative transcriptomic and metabonomic analyses reveal the regulatory effect and mechanism of tea extracts on the biosynthesis of Monascus pigments” presents the investigation on the investigate the regulatory effect and mechanism of tea extracts (rich in polyphenols) on the biosynthesis of MPs by comparative transcriptomic and metabonomic analyses, combined with RT-qPCR. The paper is good written, data are well presented, discussion is appropriate. However, due to lack of some information, in my opinion, presented manuscript needs major revision.
My major comments are presented below.
Major concerns:
- Abstract – what is the novelty of the presented manuscript
- Abstarct/Introduction – clearly define, why the presented topic is important
- Abstract/Introduction – what is the novelty of the used methods in the presented study
- Introduction is too long and should presents most important aspects of the study
- Materials and methods section – what was the sample storage protocol? Did the Authors identify any influence of sample storage conditions on the obtained results?
- Materials and methods – GC-TOF-MS section – was the ion source energy (70 eV) optimized?
- Figure 2C – due to the low size of panel C and lots of data, the quality of figure is low. I understand the idea of Authors in such way of data presentation but it is not clearly visible.
- check English and correct errors, check style and correct, remove errors in the text
Make changes in the text.
Check and correct English
Author Response
Thank you very much for your comments, which are very helpful to improve the quality of our manuscript. The specific reply is shown in the attachment. We have also uploaded a reply on the website for your review.
Review expert 4
Manuscript proposed by Li and co-workers (foods-1902255) entitled “Comparative transcriptomic and metabonomic analyses reveal the regulatory effect and mechanism of tea extracts on the biosynthesis of Monascus pigments” presents the investigation on the investigate the regulatory effect and mechanism of tea extracts (rich in polyphenols) on the biosynthesis of MPs by comparative transcriptomic and metabonomic analyses, combined with RT-qPCR. The paper is good written, data are well presented, discussion is appropriate. However, due to lack of some information, in my opinion, presented manuscript needs major revision.
My major comments are presented below.
Major concerns:
Q1- Abstract – what is the novelty of the presented manuscript
A1: Thanks for your constructive comment. We have revised the abstract and briefly mentioned the novelty of this work in the abstract.
Q2- Abstarct/Introduction – clearly define, why the presented topic is important
A2: Thanks for your comment, which is very helpful to us. We noticed that we did not explain why we want to increase the production of MPs to reflect the importance of the topic. Thank you for your comment, which is very helpful to us. Correspondingly, we explained in the introduction that the demand of MPs is increasing gradually, and it still mainly depends on biological fermentation for production, which shows the importance of this topic. Thank you again for your advice.
Q3- Abstract/Introduction – what is the novelty of the used methods in the presented study
A3: Thanks for your constructive comment. We have revised the abstract and introduction, and pointed out the novelty of the research method of this work briefly.
Q4- Introduction is too long and should presents most important aspects of the study
A4: Thanks for your comment. We agree with you, so we have simplified the content of the introduction and focused on the research progress related to our research, and introduced the characteristics of research methods and main content of this work.
Q5- Materials and methods section – what was the sample storage protocol? Did the Authors identify any influence of sample storage conditions on the obtained results?
A5: Strains are usually stored at 4℃ after activation for short-term use, whereas tea extract samples are stored at -20℃. Sequencing samples and metabolomics samples were stored at -80℃ and sent for testing after fermentation. I am sorry that we did not verify the effect of storage conditions on the results, because the fermentation period was not long and all samples were stored at low temperature in a timely manner.
Q6- Materials and methods – GC-TOF-MS section – was the ion source energy (70 eV) optimized?
A6: Thanks for your insightful comment. Our metabolomics test program mainly refers to Huang, Z B. with some simple modifications of the heating program, but we are sorry that we haven't optimized ion source energy.
Reference:
Huang, Z.B.; Zhang, S.Y.; Xu, Y.; et al. Metabolic effects of the pksCT gene on Monascus aurantiacus Li As3.4384 using Gas Chromatography-Time-of-Flight-Mass Spectrometry Based metabolomics. J Agric Food Chem 2016; 64(7): 1565-1574.
Q7- Figure 2C – due to the low size of panel C and lots of data, the quality of figure is low. I understand the idea of Authors in such way of data presentation but it is not clearly visible.
A7: Thanks for your comment. We are sorry for the poor reading experience caused by the unclear picture. To improve this situation, we adjusted Figure 2 accordingly. Firstly, Hierarchical Cluster plot is transferred to the supplementary file to give larger display space to Figure 2B and Figure 2C. Second, we have enlarged the fonts in Figures 2A and 2C for clearer viewing.
Q8- check English and correct errors, check style and correct, remove errors in the text
-Make changes in the text.Check and correct English
A8: Thanks for your comment. After inspection, we did find some clerical errors and inconsistencies in the form of proper nouns, and we have corrected these errors accordingly. Besides, we have also polished some English expressions in order to improve readers' reading experience.

Round 2
Reviewer 4 Report
The revised version of the presented manuscript meets all of my requirements. Authors gave answers, the text was modified according to my comments.